# Analyzing Changes in Attitudes and Behaviors towards Seasonal Influenza Vaccination in Spain’s Adult Population over Three Seasons

**DOI:** 10.3390/vaccines12101162

**Published:** 2024-10-12

**Authors:** Camino Prada-García, Marina Toquero-Asensio, Virginia Fernández-Espinilla, Cristina Hernán-García, Iván Sanz-Muñoz, Jose M. Eiros, Javier Castrodeza-Sanz

**Affiliations:** 1Department of Preventive Medicine and Public Health, University of Valladolid, 47005 Valladolid, Spain; vfernandeze@saludcastillayleon.es (V.F.-E.); chernang@saludcastillayleon.es (C.H.-G.); jjcastrodeza@saludcastillayleon.es (J.C.-S.); 2National Influenza Centre, Edificio Rondilla, Hospital Clínico Universitario de Valladolid, 47009 Valladolid, Spain; mtoquero@icscyl.com (M.T.-A.); isanzm@saludcastillayleon.es (I.S.-M.); jmeiros@uva.es (J.M.E.); 3Dermatology Service, Complejo Asistencial Universitario de León, 24008 León, Spain; 4Preventive Medicine and Public Health Service, Hospital Clínico Universitario de Valladolid, 47003 Valladolid, Spain; 5Instituto de Estudios de Ciencias de la Salud de Castilla y León (ICSCYL), 42002 Soria, Spain; 6Centro de Investigación Biomédica en Red de Enfermedades Infecciosas (CIBERINFECC), 28029 Madrid, Spain; 7Microbiology Service, Hospital Universitario Río Hortega, 47012 Valladolid, Spain

**Keywords:** vaccine hesitancy, healthcare communication, attitudes, behaviors, public health

## Abstract

**Background/Objectives**: The experience of the COVID-19 pandemic has turned the spotlight on the importance of public health measures and disease prevention. Despite this, the acceptance of influenza vaccination has remained low in most countries (and far from the 75% target set by the World Health Organization). The objective of this study has been to investigate how attitudes and behaviors regarding influenza vaccination in the Spanish adult population have changed over the last three years (from 2021 to 2024) in order to analyze trends in influenza vaccination. **Methods**: To this end, a cross-sectional study was conducted through 2206 telephone interviews, and the results were compared with those obtained in previous campaigns. **Results**: The findings indicate a significant decline in overall vaccination intent. Healthcare professionals remain the most influential factor in encouraging vaccination, yet there is a notable increase in the lack of vaccine recommendations, contributing to the decision not to vaccinate. This study also reveals low awareness of the influenza vaccine campaign, emphasizing the need for improved public health communication. **Conclusions**: To counteract these trends, this study recommends intensifying awareness campaigns, strengthening the role of healthcare providers in vaccine advocacy, and tailoring communication strategies. These efforts are crucial to enhancing vaccination coverage and protecting vulnerable populations against influenza.

## 1. Introduction

Influenza is a highly infectious respiratory disease characterized by the acute onset of symptoms, including fever and cough, which can lead to severe complications and even death. It causes a considerable disease burden in terms of excessive morbidity, mortality, and hospitalization [1,2,3,4]. Moreover, the disease poses an additional load on the healthcare system and increases the economic burden as a result of work absenteeism and loss of productivity [5,6,7]. Vaccination is one of the most significant, successful, and cost-effective public health measures [8,9]. The primary aim of influenza vaccination is to protect high-risk groups [10,11,12]; however, vaccine acceptance in most European countries remains below the 75% target level in people aged 65 and older set by the World Health Organization and the European Centre for Disease Prevention and Control (ECDC) [13,14,15]. In Spain, the influenza vaccination coverage in the year 2023 was 65.97% in people aged 65 and older [16].

In recent years, there has been a strong focus on identifying potential barriers to vaccine acceptance. In 2019, the World Health Organization (WHO) officially defined vaccine hesitancy as the delay or refusal to vaccinate due to a lack of knowledge about vaccine safety, effectiveness, and the diseases they protect against. This categorization placed it among the top 10 global health threats [17,18]. Understanding the obstacles to vaccination and the factors that encourage vaccine uptake is essential for enhancing influenza vaccination coverage. Vaccine acceptance is influenced by a variety of factors, including personal beliefs, societal influences, and logistical issues [19]. Moreover, research shows that elements such as age, gender, access to healthcare, education, income, socioeconomic status, and the presence of chronic diseases significantly impact vaccine uptake, particularly among the elderly [20,21,22]. During the COVID-19 pandemic, some studies evaluated the population’s intentions to receive the influenza vaccine [14,23,24,25,26,27,28,29]. These studies indicated that the pandemic created a shift in public attitudes towards vaccines in general, often increasing awareness of the importance of immunization, but also introducing new barriers such as vaccine fatigue.

The COVID-19 pandemic has significantly influenced public perceptions of vaccination. The lifting of the Public Health Emergency of International Concern (PHEIC) status by the World Health Organization (WHO) in 2023 has shifted public health priorities, yet COVID-19 virus strains continue to circulate and, in some settings, lead to new increases in hospitalization and intensive care unit admissions [30]. Several studies have suggested that the heightened attention to public health during the pandemic has had both positive and negative effects on attitudes toward influenza vaccination. On the one hand, the pandemic underscored the critical role of vaccination in preventing disease spread; on the other hand, the phenomenon of vaccine fatigue, defined as “a person’s inertia or inaction towards vaccine information or instruction due to perceived burden and burnout” [31,32], has posed a challenge for maintaining high vaccination rates.

The primary objective of this study was to investigate changes in attitudes and behaviors regarding influenza vaccination in the Spanish adult population over the last three years in order to analyze trends and predictors of influenza vaccination. As our dataset spans several influenza seasons, specifically 2021/22, 2022/23, and 2023/24, we also wanted to study whether public confidence in influenza vaccination has seen changes over time. Addressing vaccine hesitancy is a public health priority, and knowing the evolution of these trends over time will help support the future development of precise intervention measures as well as focused education and training.

## 2. Materials and Methods

### 2.1. Study Design

During the 2023/24 season, a comprehensive national cross-sectional observational study was undertaken, utilizing an anonymous questionnaire that was designed similarly to those employed in the studies from 2021 and 2022 [14,25]. The primary goal of this survey was to gather detailed insights into the opinions, attitudes, and behaviors of Spanish adults concerning influenza vaccination as of September 2023. This survey was meticulously designed to comply with the General Data Protection Regulation (GDPR) guidelines, thereby ensuring the confidentiality and protection of respondents’ personal information. Participation in the survey was entirely voluntary, and the implementation of this study was overseen by GAD3, a professional firm that ensured the proper execution and integrity of the data collection process.

Following the data collection, a comparative analysis was performed, integrating the results obtained from the 2021/22 and 2022/23 seasons. This involved conducting a repeated cross-sectional study to meticulously analyze the shifts and trends in the opinions of Spain’s adult population regarding influenza vaccination over the span of three seasons. The objective of this extended analysis was not only to describe these trends but also to identify key predictors and factors influencing vaccination behaviors and attitudes. By doing so, this study aimed to provide a comprehensive understanding of the dynamics of influenza vaccination acceptance and resistance among Spanish adults, offering valuable insights into future public health strategies and interventions.

### 2.2. Target Population and Sampling Method

The research encompassed all Autonomous Communities within its geographical scope, targeting individuals aged 18 and older. This study was carried out by GAD3, a consultancy specializing in social research and communication [33] in partnership with CSL Seqirus. Data collection was conducted using Computer-Assisted Telephone Interviewing (CATI). To ensure a thorough understanding of this study’s objectives and methodology, the interviewers underwent extensive training. This training included a comprehensive group video call, during which this study’s analysts provided context for the research, explained the details of the questionnaire, and offered specific technical guidance on how to deliver questions and accurately code responses. Throughout the data collection period, the technical team maintained constant contact with the interviewers to provide ongoing support and ensure consistency in the data collection process.

Importantly, no incentives were offered to participants to encourage their involvement in this study. A non-probability quota sampling method was used, focusing on quotas based on sex, age, and geographic distribution. This approach was implemented to ensure that the sample was representative of the diverse population across all Autonomous Communities in Spain. The selection process was guided by these quotas, ensuring demographic representativeness rather than being a purely random sample. The sample size consisted of 2206 randomly selected interviews, which were evenly divided between landline and mobile phone users. These interviews were conducted over a period from 1 September to 20 September 2023, between the hours of 9 a.m. and 9 p.m.

A homogeneity analysis was conducted to evaluate the consistency of sociodemographic and risk group characteristics across the survey years (2021, 2022, and 2023). This analysis revealed that gender and age distributions remained stable, indicating a robust basis for temporal comparisons. However, differences in education level and specific risk groups (such as diabetics and healthcare professionals) suggest some variability that should be taken into account when interpreting subgroup analyses.

Prior to initiating the survey, participants were contacted by telephone and provided with a detailed explanation of this study’s purpose and their rights as participants. Verbal informed consent was obtained from each participant before including them in the survey. Participants were informed that their participation was voluntary and that they could withdraw at any point without any consequences. This procedure was designed to comply with ethical standards and ensure that participants were fully aware of this study’s purpose and their involvement.

We recognize that a stratified sampling approach by age, sex, and geographic clusters would have been ideal for ensuring greater representativeness. However, due to logistical and resource constraints, this was not feasible across all Autonomous Communities in Spain. Instead, we employed a non-probability quota sampling method, focusing on key demographic characteristics (age, sex, and geography) to approximate representativeness while maintaining practicality. Although non-probability quota sampling does not strictly require a large sample size, we chose to collect data from 2206 individuals to enhance the reliability of subgroup analyses and provide a more comprehensive overview of the population. This decision ensured sufficient statistical power to detect meaningful patterns within different demographic groups, thereby improving the validity of our findings. The questionnaire used in this study included 20 closed-ended questions, designed to gather comprehensive data within a relatively short interview duration of approximately 4 to 5 min. This approach was consistent with the methodology employed in similar studies conducted in previous years [14,25], ensuring comparability of the results over time.

Participants eligible for this study were individuals residing in Spain, aged 18 or above, who voluntarily consented to take part. Those not meeting these conditions were excluded. The sampling was conducted using quotas for sex, age, and geographic distribution, employing a non-proportional allocation technique.

This careful data collection and participant selection approach aimed to provide a deep and comprehensive insight into the attitudes and behaviors of the population in Spain concerning influenza vaccination. By ensuring that this study encompassed all Autonomous Communities and included a representative sample of individuals aged 18 and older, the research aimed to capture the diverse perspectives and experiences across different regions and demographics. The random sampling method and the voluntary nature of participation further ensured that the findings would be unbiased and reflective of the general population. This robust methodology provided a solid foundation for analyzing variations and trends, ultimately contributing to more targeted and effective public health strategies.

### 2.3. Questionnaire

The original questionnaire from 2021 was slightly modified [25]. Demographic variables such as age, gender, current employment status, education level, belonging to risk groups, and postal code of residence were recorded. Additionally, vaccination status in previous years was documented, along with the most relevant reasons for deciding to be vaccinated or not. The degree of information received about the vaccination campaign, as well as the means through which the participant received information, were also noted. In this study, ‘attitudes’ refer to the general beliefs and perceptions about influenza vaccination, which were assessed through questions evaluating participants’ perceived benefits, risks, and overall trust in the vaccine. ‘Acceptance’ refers to whether participants had previously received the influenza vaccine or were willing to receive it under certain circumstances. This was measured by questions about participants’ past vaccination behavior and their current openness to vaccination. ‘Intentions’ refer to the participants’ stated plans to get vaccinated in the upcoming influenza season, assessed through direct questions on their willingness to receive the vaccine during the next campaign. Each of these constructs was measured using distinct items within the survey to ensure clarity and accuracy in capturing the different aspects of vaccine behavior. This study also recorded the intention to receive the flu vaccine this year, along with the key reasons for choosing to get vaccinated or not. Additionally, it included the perceived importance of vaccination compared to previous years, the necessity for a specific flu vaccine for older adults, potential actions to promote vaccination, preferred vaccination sites, the significance of flu vaccines in the current context, and the coadministration with the COVID-19 vaccine.

This comprehensive approach aimed to gather detailed insights into the multifaceted aspects of influenza vaccination, providing valuable data for designing targeted public health strategies.

### 2.4. Statistical Analysis

Although a non-probability quota sampling approach was employed, we aimed to ensure that the sample size would be large enough to provide reliable estimates across different demographic groups and enhance the robustness of the analysis. To communicate the degree of reliability we aimed to achieve, we calculated an approximate margin of error of ±2.1%, assuming a conservative scenario where P = Q = 0.5. This calculation was based on the binary variable representing vaccination status (yes/no), and P = Q = 0.5 was chosen to represent the scenario with maximum variability, which provides the most conservative estimate for the required sample size. While these calculations are more typical for probability-based sampling, our intention was to enhance the robustness of our non-probabilistic sample and approximate representativeness as closely as possible.

To analyze the data, descriptive statistics were used to summarize the demographic characteristics of the sample. In addition, logistic regression analysis was conducted to assess the relationship between respondents’ willingness to be vaccinated in this vaccination campaign and key predictors such as age, underlying diseases, and whether they had been previously vaccinated. Chi-square tests were also conducted to compare categorical variables across different groups, such as vaccination uptake by gender and age group.

For the multivariate logistic regression analysis, the outcome variable was the response to the following query: “*Considering the*
*upcoming flu vaccination campaign, do you plan to get vaccinated?*”. The predictor variables included in the model were as follows:•Being part of a risk group.•Age categorized into two groups (65 years or older, under 65).•Binary response (Yes or No) to the following question: “*In the past few years, have you received a flu vaccination?*”

The variables were introduced into the model based on their statistical significance, and the R^2^ value was tracked to identify the best-fitting model. No goodness-of-fit test was applied to the model. The odds ratios (OR) for the studied variables were computed using logistic regression, adjusted for the selected predictors, with 95% confidence intervals to assess the strength of the associations. The threshold for statistical significance was set at *p* < 0.05, indicating that results with a *p*-value below this threshold were considered statistically significant.

Data analysis was performed using IBM SPSS Statistics version 27 (IBM Corp, Armonk, NY, USA) and the R Project for Statistical Computing for Windows. This comprehensive analytical approach helped identify key predictors and their influence on the likelihood of intending to get vaccinated against influenza.

## 3. Results

The findings of this study are presented in three main sections: the first section details the sociodemographic variables considered; the second explores changes in the population’s attitudes and behaviors toward influenza vaccination across three seasons (2021/22, 2022/23, and 2023/24); and the third highlights the primary information sources and strategies used to promote vaccination.

### 3.1. Sociodemographic Variables of This Study

Of the 2206 interviews conducted, 47.7% of the participants were men and 52.3% were women. Additionally, 26.4% were aged 65 or older. In terms of employment status, 33.9% were employed in the private sector, nearly 28% were retired, and 13.8% were employed in the public sector. When it comes to education level, almost 75% had completed either secondary or university education.

Regarding risk group membership, 33.5% of respondents indicated that they belonged to a risk group. Among these, 60% were individuals over 60 years old, and 18.9% had some form of respiratory condition.

As seen, the distribution of sociodemographic variables in this third survey was very similar to the values obtained in the two previous seasons [14,25] (Table 1). An analysis of homogeneity was conducted to assess the stability of sociodemographic characteristics across the survey years (2021, 2022, and 2023). This analysis confirmed that the gender and age distributions remained stable, providing a solid foundation for temporal comparisons. However, significant differences were noted in education level and certain risk groups (e.g., diabetics and healthcare professionals), suggesting variability that must be considered when interpreting subgroup results.

### 3.2. Changes in Attitudes and Behaviors Regarding Influenza Vaccination

The analysis of past vaccination rates indicated an increase over the last year, showing that over a third of Spanish residents received the flu vaccine each year (34.4%), while 48.3% reported never having been vaccinated against influenza. The Community of Madrid emerged as the region with the highest annual vaccination rates. Among individuals aged 60 or above, 66% were vaccinated annually; for those aged 65 or older, the rate was 72.7% (similar to the previous season); and up to 84.4% reported being vaccinated annually, frequently, or occasionally, compared to 39.7% of individuals under 65 years (*p* < 0.001). Additionally, up to 80% of citizens in a risk group were vaccinated annually, frequently, or occasionally, compared to 36.7% of those not in a risk group (*p* < 0.001), a rate comparable to the previous season (Table 2). The Chi-square test was employed to compare vaccination rates across different demographic groups, including age categories and risk group status.

In our study, we compared responses from the 2023/24 season to those from 2022/23 across different groups: individuals in risk groups, those not at risk, participants under 65, those aged 65 and above, as well as the general population. The analysis indicated no statistically significant differences between the two seasons in attitudes and behaviors regarding prior influenza vaccination rates, either for the general population or for any of the specific groups mentioned.

The primary reasons for getting vaccinated against influenza among participants were personal and environmental protection (62.1%) and healthcare provider recommendations (57.3%). Furthermore, 19.3% cited social responsibility as their reason and 9.7% mentioned vaccination for both influenza and COVID-19, with a notable decline in these two reasons compared to the previous season. When analyzed by age, the motivations “because I have sufficient information about the vaccine”, “due to my doctor’s recommendation”, and “based on a nurse’s advice” were more commonly cited by participants aged 65 or older compared to younger individuals (*p* < 0.05). On the other hand, younger participants (under 65) more frequently mentioned social responsibility and protection for themselves or their environment (*p* < 0.05). Comparing risk groups, reasons such as “for my own and my environment’s protection”, “having experienced the consequences of influenza before”, “social responsibility”, “sufficient knowledge about the vaccine”, “doctor’s recommendation”, and “nurse’s advice” were more often reported by those in risk groups (*p* < 0.05).

The Chi-square test was used to compare the reasons for deciding to vaccinate or not between different demographic groups, such as age and risk group membership. Compared to the previous season, there was a significant decrease (*p* < 0.001) for all reasons except “because my doctor recommended it” (Table 3). Additionally, these differences were also found among people aged 65 or older.

A total of 66.3% of Spanish participants who had not received the influenza vaccine in recent years indicated that it was not recommended or prescribed to them, which marks a 14-point increase compared to the previous season (*p* < 0.001) and nearly a 28-point rise since the 2021 survey. Among individuals aged 65 and above, the percentage remained stable at 47.8%, similar to the prior season. Additionally, 39.1% of respondents believed influenza was not severe, showing an increase of 7 points compared to last year’s survey (*p* < 0.001). There was also a significant reduction in the percentage of participants lacking information about the vaccine, from 14.9% to 6.0% (*p* < 0.001), in those feeling unwell following prior vaccination, decreasing from 6.9% to 4.5% (*p* = 0.024), and in needle phobia, dropping from 5.4% to 2.1% (*p* < 0.001) (Table 3). These differences were not significant among individuals aged 65 or older. Based on age, reasons such as “lack of confidence in the vaccine’s effectiveness” and “previous vaccination made me feel ill” were more often mentioned by those aged 65 and above compared to younger respondents (*p* < 0.05). On the other hand, the reason “not recommended/prescribed” was more frequently cited by those under 65 (*p* < 0.001). When considering risk group status, reasons like “needle phobia” and “previous vaccination made me feel ill” were more commonly reported by individuals belonging to a risk group (*p* < 0.05).

Compared to previous years, the intention to get vaccinated during the 2023/24 influenza season has dropped to 44.4%, with more people indicating that they do not plan to get vaccinated in the near future (51.8%). It is the older individuals and the rest of the risk groups who stand out in their desire to get vaccinated. In this way, among those aged 65 or older, this figure increased to 80.6%, a percentage slightly lower than the previous season, although without showing significant differences. For those aged 60 or older, the percentage was 74%. In contrast, only 33.7% of individuals under 65 expressed the same intention (*p* < 0.001), almost 6 points less than in the previous season (*p* < 0.05). The Community of Madrid and Extremadura stand out as the areas where around half of the population intends to get vaccinated, showing the highest intention to receive the influenza vaccine, with 53% and 50% respectively. Furthermore, 81.2% of Spaniards who belong to a risk group intended to get vaccinated during the 2023/24 season, which is 2 percentage points lower than the previous year, compared to 28.7% of those not in a risk group (*p* < 0.001). Relative to the previous season, this percentage dropped by almost 9 points (*p* < 0.001).

A total of 98% of individuals who received the vaccine consistently each year stated they intended to get vaccinated again for the 2023/24 season. Among those who received the vaccine frequently or occasionally, 69% and 35%, respectively, planned to do so. On the other hand, only 7% of participants who had never previously been vaccinated against influenza expressed an intention to get vaccinated in the current campaign, which is 8% lower compared to the previous season (*p* < 0.001) and 12% lower than during the 2021/22 season. Regarding employment status, 77.6% of retirees planned to get vaccinated, in contrast to 19.5% of students and 23.0% of self-employed individuals or entrepreneurs (*p* < 0.001). Compared to the previous season, vaccination intention was lower across all employment statuses except among domestic workers. As observed in prior campaigns, individuals with only primary education showed a greater intention to get vaccinated compared to those with higher education (*p* < 0.001) (Table 4). Chi-square tests were utilized to assess differences in vaccination intentions across various categories, such as age, risk group status, and employment status.

As in the previous two years, the need to protect oneself against the virus (57.3%) and the practice of getting vaccinated annually (49.3%) were the two main reasons why the Spanish population intended to get vaccinated, with a decreasing trend observed for both reasons (*p* < 0.05). Social responsibility was the third most common reason for getting vaccinated (35.3%), showing a decrease of 8 percentage points compared to the previous season (*p* < 0.001). Trust in vaccines was cited by 27.0% of participants, reflecting a 15-point drop compared to the prior year (*p* < 0.001). The motivations “to get vaccinated against both influenza and COVID-19 simultaneously” (6.8%) and “COVID-19 has made me more aware of the importance of vaccination” (5.9%) both experienced a 20-point decline compared to last year, indicating a reduced influence of COVID-19 on influenza vaccination decisions (*p* < 0.001). Regarding age, reasons such as “the need for an annual flu vaccine” and “to receive influenza and COVID-19 vaccines at the same time” were mentioned more often by participants aged 65 and above compared to younger groups (*p* < 0.001). In contrast, those under 65 years old cited greater confidence in vaccines (*p* < 0.05) and a stronger sense of social responsibility (*p* < 0.001). Compared to the responses from last year in the group of people aged 65 or older, significant differences were found in the reasons “you need to protect yourself against viruses”, which dropped from 72.5% to 56.1%, “I trust vaccines in general”, which decreased from 43.7% to 23.9%, “because of social responsibility”, which fell from 36.1% to 21.5%, “because COVID-19 has made me more aware of the importance of vaccination”, which dropped from 17.6% to 4.8%, and “to be vaccinated against influenza and COVID-19 at the same time”, which decreased from 26.8% to 9.9% (*p* < 0.001). In terms of belonging to risk groups, the most commonly cited reasons among those in these groups were the same as in the previous season: the necessity to protect against viruses (*p* < 0.05), the need for an annual influenza vaccination (*p* < 0.001), and to be vaccinated against both flu and COVID-19 simultaneously (*p* < 0.05).

Conversely, for those who choose not to get vaccinated, the reasons for lack of recommendation or prescription (54.8%) and the perception of influenza as not being severe (32.0%) have increased compared to the previous season, following an upward trend. These two reasons continue to be the most significant factors influencing the decision not to get vaccinated. A bad experience after getting vaccinated in the past has emerged as a new reason among respondents (7.7%) (Table 5). Focusing on age, the lack of trust in vaccines was mentioned more frequently among people aged 65 or older (14.2%) than among those under 65 years (7.2%) (*p* < 0.05), as well as the vaccine’s ineffectiveness (10.3% vs. 2.5%) (*p* < 0.001) and a bad experience after getting vaccinated in the past (17.3% vs. 6.6%) (*p* < 0.001). On the other hand, younger individuals more often cited the lack of recommendation or prescription (57.1%) compared to those 65 or older (34.6%) (*p* < 0.001). The reason “I got vaccinated once, and it made me feel sick” was more frequently given by people belonging to a risk group (12.7% vs. 6.9%) (*p* < 0.05). The Chi-square test was employed to evaluate the differences in reasons for vaccine hesitancy or refusal between different demographic groups, such as age and risk group membership.

Regarding vaccination location preferences, their Health Center remains the preferred location for Spaniards to receive their vaccinations (86.7%), with an increased preference for workplace vaccination compared to the previous year (12.4%). Meanwhile, home vaccination, particularly in cases of mobility issues, and hospital vaccination have seen a slight decline, with preferences at 8.5% and 5.2%, respectively. In terms of age differences, individuals aged 65 and above showed a stronger preference for their Health Center (*p* < 0.001), just like in the previous campaign, whereas younger people were more inclined to choose hospitals (*p* < 0.05), specific vaccination sites, similar to those used for COVID-19 (*p* < 0.001) and workplaces (*p* < 0.001).

The importance of the influenza vaccine is, for more than half of Spaniards (56.5%), the same as in previous years. Only 13.8% of Spaniards are aware of the existence of a specific vaccine for the elderly, with this awareness being slightly higher than average among younger people (20%). The perception of the need to acquire a vaccine specifically for the elderly has increased compared to previous years, with more than two-thirds of the population (68%) now considering it necessary under any circumstances. Those under 65 years more frequently indicated than those aged 65 or older that they were aware of the existence of a specific vaccine for the elderly and that the Administration should acquire it. For 4 out of 10 Spaniards, the simultaneous administration of influenza and COVID-19 vaccines would be considered an improvement. On the other hand, more people would be willing to get vaccinated only against influenza compared to COVID-19 (14.5% and 7.4%, respectively). The belief in the benefits of joint vaccination increases as the age of the respondents rises.

In our logistic regression models, we analyzed various variables to identify predictors of vaccination intent for the 2023/24 campaign. The model that provided the best fit, with an R² of 0.521, indicated that age, risk group membership, and previous vaccination history were significant predictors. Specifically, individuals aged 65 and above were twice as likely to intend to get vaccinated compared to younger participants, while those belonging to a risk group were 5.4 times more likely than those not in a risk group. Furthermore, individuals with a history of previous vaccination—whether annually, frequently, or occasionally—were 41.4 times more likely to intend to get vaccinated during the 2023/24 campaign (Table 6).

### 3.3. Information and Measures to Promote Vaccination

The publicity surrounding the influenza vaccination campaign remains relatively unknown among Spaniards, with only 15.8% claiming to have received any information this year. Additionally, among individuals aged 65 or older, the figures remained at 16.8%. Among those who reported receiving any information about the vaccination campaign, the traditional media was the main source of knowledge (39%), followed by the Public Administration (24%), which has increased as a primary source of information compared to previous years. Healthcare professionals slightly decreased (19.2%), now occupying the third place, unlike the past two years when they were the main source of information. Moreover, 6.6% of participants reported obtaining information at their workplace (Figure 1). For those aged 65 and older, traditional media was the main source of information (51%), while healthcare providers declined from 44.2% last year to 17.3% in the current season. It was also noted that social media continued to be used at very low rates (1%) among this group (*p* < 0.05). Additionally, 27.6% of individuals in risk groups reported receiving information about the vaccination campaign from healthcare professionals, in contrast to 13.4% of those not considered at risk. The traditional media was the main source of information among Andalusians, Galicians, and residents of Madrid. However, among people from Murcia, the Public Administration and healthcare professionals play a more significant role as the primary sources of information (26%).

The recommendation from healthcare professionals was the most influential factor in encouraging the Spanish population to get vaccinated (85.8%), followed by increased awareness of the vaccine (81.2%) and SMS (Short Message Service) reminders (67.9%). Only 52.8% of respondents believed that so-called “vaccination hubs” were effective in promoting vaccination. Compared to the previous season, awareness campaigns in the media have seen a reduced capacity to motivate vaccination (*p* < 0.001) (Table 7).

## 4. Discussion

Influenza vaccination is one of the most effective public health measures to reduce disease burden, especially in vulnerable populations such as the elderly and those with comorbidities [36]. However, vaccine acceptance remains a challenge in many countries, including Spain, where coverage rates have fluctuated in recent years. The COVID-19 pandemic has had a significant impact on public perceptions of vaccines in general, including influenza vaccines, influencing both vaccination intent and confidence in the effectiveness of these interventions [37,38,39]. Although influenza vaccination did not reach the priority levels of the COVID-19 vaccine, increased attention and interest in vaccination were observed during the peaks of the pandemic [28].

Previous studies have shown that the recommendation from healthcare professionals is a critical factor in the decision to get vaccinated [14,25,27,40,41]. However, factors such as vaccine fatigue, diminished risk perception, and variable confidence in vaccines have negatively impacted vaccination rates in recent years [42]. These aspects are particularly relevant in the context of the 2023/24 season, where, despite the stabilization of vaccination rates in high-risk groups, a general decrease in vaccination intent has been observed.

This study was conducted to investigate the attitudes and behaviors of the Spanish adult population regarding influenza vaccination over the past three years, specifically from the 2021/22 to the 2023/24 seasons. By analyzing these trends, this study aimed to identify the key factors that have influenced vaccination behavior during this period, including the impact of the COVID-19 pandemic and subsequent public health responses. Understanding these evolving trends is crucial for informing future public health strategies, particularly in enhancing vaccine uptake and addressing hesitancy within different segments of the population. This study also sought to explore how confidence in the influenza vaccine has changed over time among different demographic groups.

### 4.1. Perspectives and Behaviors Concerning Influenza Vaccination

In the 2023/24 season, 34.4% of people in Spain got vaccinated against influenza, while 48.3% remained unvaccinated. Although there is a slight increase in uptake, nearly half of the population not being vaccinated presents a significant public health challenge, consistent with trends seen in other European countries due to vaccine hesitancy and logistical barriers [19].

Madrid led in vaccination rates, likely due to effective local campaigns and healthcare access [1]. Older age continues to be a strong predictor of vaccination, with 66% of individuals aged 60+ and 72.7% of those aged 65+ vaccinated, compared to 39.7% of those under 65 (*p* < 0.001). Vulnerable populations also show higher uptake (80% for risk groups vs. 36.7% for non-risk groups, *p* < 0.001), a trend consistent with findings on vaccine acceptance among those at higher risk [2].

Key motivators for vaccination were protection and healthcare provider recommendations, aligning with [19]. However, motivations like social responsibility have decreased, likely due to pandemic fatigue and reduced perceived urgency as noted by [43]. Older adults valued healthcare recommendations and information more, highlighting the effectiveness of targeted campaigns for this group [44].

Compared to the 2022/23 season, there was a significant decline in most reasons for getting vaccinated, with the exception of doctor recommendations. This decline may reflect an increase in “vaccine fatigue” as the perceived risk of influenza decreases with the receding of the COVID-19 pandemic. Notably, even among individuals aged 65 or older—traditionally a group with high vaccination intent—this decline is evident.

A major concern identified is the lack of vaccination recommendations by healthcare professionals. In 2023/24, 66.3% of unvaccinated Spaniards reported that the vaccine was neither recommended nor prescribed, a 14-point increase from the previous year and nearly 28 points higher compared to 2021 (*p* < 0.001). This trend indicates a potential decline in proactive healthcare communication and raises issues of vaccine hesitancy among healthcare providers. Previous studies have shown that healthcare professionals’ own vaccination behaviors are directly linked to their willingness to recommend vaccines to patients, affecting overall community vaccination rates [14].

Furthermore, 39.1% of respondents perceived influenza as not severe, a 7-point increase from the prior year (*p* < 0.001). This diminished risk perception likely stems from a reduced focus on influenza now that the immediate threat of COVID-19 has lessened, thereby undermining vaccination efforts, especially among those who underestimate the potential severity of influenza.

On a positive note, fewer respondents cited a lack of information about the vaccine (dropping from 14.9% to 6.0%, *p* < 0.001), and concerns about feeling ill after vaccination or fear of needles also decreased (*p* = 0.024 and *p* < 0.001, respectively). These improvements suggest that public health campaigns have effectively reduced misinformation and alleviated fears about vaccination side effects.

However, age-related vaccine hesitancy persists. Older adults (65 and older) were more likely to report a lack of confidence in vaccine effectiveness and cite negative past experiences compared to younger individuals (*p* < 0.05). Despite being at higher risk, this age group appears to be influenced by past negative experiences and skepticism, indicating the need for targeted communication addressing these specific concerns.

This study identifies several challenges already acknowledged in public health practice, such as the crucial role of healthcare professionals in promoting influenza vaccination and the persistent issue of low vaccination coverage among high-risk groups. These findings align with previous research but provide updated data specific to the Spanish population in the post-COVID-19 context. Moreover, this study highlights emerging issues that have not been extensively addressed before, such as the rising trend of ‘vaccine fatigue’ and a declining perceived severity of influenza, both of which are contributing to reduced vaccination intent. Additionally, the increasing lack of proactive vaccine recommendations from healthcare professionals presents a novel challenge that requires urgent attention. These insights contribute valuable knowledge that can inform targeted strategies to enhance vaccination uptake, especially through improved communication and engagement from healthcare providers.

Vaccination intent dropped to 44.4% in 2023/24, with over half indicating they would not get vaccinated. This trend of complacency echoes concerns noted by [45], which suggests that declining perceived risk after the pandemic can severely impact vaccination campaigns. Young and working-age groups showed lower intent, likely due to perceived lower risk and logistical barriers, in line with studies highlighting the need for targeted interventions to increase uptake among these populations [22].

Motivations such as self-protection and annual vaccination declined (*p* < 0.05), while social responsibility dropped significantly by 8 points (*p* < 0.001). This trend may reflect reduced urgency as COVID-19 fades from public focus, similar to observations made by the authors of [43] about pandemic fatigue leading to reduced engagement. Trust in vaccines fell sharply by 15 points (*p* < 0.001), suggesting lingering impacts of pandemic-related hesitancy, which could hinder not only influenza vaccination but also other public health efforts [14].

The influence of COVID-19 on influenza vaccination has declined significantly, with motivations related to co-infection prevention dropping sharply. As the urgency of the pandemic fades, its impact on promoting other vaccinations diminishes as well, which aligns with findings from the authors of [43] about the challenge of maintaining vaccination motivation post-pandemic.

For those opting not to get vaccinated, the primary reasons were a lack of recommendation and the perception that influenza is not severe, both of which have increased. These findings highlight a persistent gap in healthcare communication, consistent with the authors of [19], who emphasize the crucial role of healthcare professionals in shaping public health behaviors.

Health Centers remain the preferred site for vaccination (86.7%), reflecting strong trust in primary healthcare. The increase in preference for workplace vaccinations (12.4%) indicates a growing demand for convenience, especially among the working population, which suggests the potential for expanding workplace vaccination programs to increase coverage. Similar trends have been noted by the authors of [22], who found that convenient access can significantly enhance vaccination rates.

Awareness of specific vaccines for the elderly remains low (13.8%), despite slightly higher awareness among younger people. This discrepancy may be due to greater engagement with digital health communication platforms among the younger demographic. The belief in the benefits of simultaneous administration of influenza and COVID-19 vaccines was more prevalent among older adults, aligning with the authors of [20], who found older populations are more receptive to combined vaccination strategies when risk perception is higher.

Across the three seasons, the logistic regression models have revealed consistency in the key predictors of vaccination intention, although with notable variations in the magnitude of their influence. In the 2023/24 campaign, a significant increase was observed in the influence of risk group membership and prior vaccination history, suggesting a growing awareness and recognition of the importance of vaccination among these groups. The intensification of the impact of prior vaccination history is particularly noteworthy, possibly reflecting stronger and more effective vaccination campaigns, as well as greater acceptance of vaccination among those with previous experience.

The evolution of these predictors suggests that as the COVID-19 pandemic continues to have residual effects, the vulnerable population is increasingly predisposed to getting vaccinated, which is a positive sign for future vaccination campaigns. However, the greater reliance on prior vaccination as a predictor also indicates the need to continue working on education and promotion to ensure that those without previous vaccination experience also feel motivated to get vaccinated.

### 4.2. Information and Measures to Promote Vaccination

The findings suggest that awareness of the influenza vaccination campaign in Spain remains low, with only 15.8% of respondents reporting any exposure to campaign information. Traditional media and public administrations were the main sources, while healthcare professionals, previously the top source, dropped to 19.2%. Particularly among those aged 65 and older, reliance on healthcare providers as an information source fell significantly from 44.2% to 17.3%. This decline may help explain lower vaccination awareness and uptake, as direct communication from trusted medical professionals is key for influencing vaccination behavior [19].

Despite the drop in campaign visibility, healthcare professionals’ recommendations still have the most significant influence, with 85.8% of respondents citing it as their motivation for vaccination. This emphasizes the enduring impact of professional guidance, aligning with the findings in [14], which highlight the crucial role of healthcare providers in vaccine promotion.

The reduced impact of media awareness campaigns may be attributed to message fatigue, suggesting that updated and more engaging communication strategies are needed to sustain public interest and motivation. This aligns with the authors of [43], who found that repetitive messaging can lead to reduced public engagement over time.

This study reinforces known predictors of vaccination, such as age, risk group membership, and healthcare recommendations, but also suggests a shift in their influence due to the COVID-19 pandemic. Trust in vaccines and awareness campaigns have shown changes in effectiveness, pointing to the need for future research into how the pandemic has reshaped these predictors, as also suggested by the authors of [44].

### 4.3. Research Limitations

Despite the valuable insights provided by this study, several limitations should be acknowledged. First, while the sample was intended to be representative at the national level, not all socioeconomic and cultural groups may have been equally represented, particularly those without access to mobile or landline telephones. Second, the data collection relied on self-reported responses from participants, which may have introduced recall bias or social desirability bias, potentially affecting the reliability of the findings. Third, although a comparative analysis across seasons was conducted, the cross-sectional design of this study limits our ability to establish causal relationships in the observed changes in vaccination attitudes and behaviors.

Additionally, the use of a non-probability quota sampling method may introduce selection bias. While quotas were employed to ensure demographic representation across gender, age, and geography, this approach lacks the randomness of probability-based sampling, which can limit the generalizability of the findings. Thus, caution should be exercised when interpreting the inferential aspects of this study, as our results may not be fully applicable to the broader population or different contexts.

Furthermore, contacting participants via landline and mobile phones may have introduced additional selection bias. This approach could lead to an underrepresentation of certain population groups, such as those without access to telephones or those who avoid answering calls from unknown numbers. This bias may affect the representativeness of socioeconomically disadvantaged groups or those with limited access to communication technologies. Future studies should consider incorporating diverse recruitment methods, such as online surveys or in-person interviews, to reduce this potential bias and ensure broader population representation.

Finally, the impact of the COVID-19 pandemic likely influenced public attitudes toward influenza vaccination in ways that may not be generalizable to a non-pandemic context, limiting the long-term applicability of the results.

## 5. Conclusions

This study provides a comprehensive overview of attitudes and behaviors toward influenza vaccination among the Spanish population during the 2023/24 season, highlighting both progress and ongoing challenges in vaccine acceptance. While there has been stabilization in vaccination rates among high-risk groups, such as individuals aged 65 and older, overall vaccination intent has significantly declined compared to previous seasons.

This study reveals a concerning drop in vaccine confidence, as well as a reduced influence of COVID-19 on influenza vaccination decisions. As the COVID-19 pandemic has become less immediate, so has its power to drive influenza vaccination. This trend underscores the need to renew communication efforts to maintain trust in vaccination, regardless of the pandemic context. Although healthcare professionals remain the most influential factor in the decision to vaccinate, this study highlights a worrying trend of an increasing lack of recommendations from these professionals. This was one of the primary reasons cited by respondents for not getting vaccinated, presenting a significant challenge for public health strategies. It is crucial that healthcare professionals proactively recommend the vaccine to counteract the growing apathy and skepticism among the population. The low level of awareness regarding the influenza vaccination campaign and the specific vaccine for the elderly points to a significant gap in public health communication. It is crucial to enhance information dissemination and ensure that messages effectively reach all segments of the population, especially the most vulnerable.

To reverse the negative trends observed and improve influenza vaccination rates, it is essential to intensify awareness campaigns and make additional efforts to strengthen vaccine confidence. This includes greater engagement from healthcare professionals in actively recommending the vaccine and the implementation of more personalized communication strategies that consider demographic and regional differences. Addressing the lack of recommendations from healthcare professionals should be a priority, as their influence is crucial for improving vaccine acceptance. Addressing these challenges will be key to ensuring better preparedness for future influenza seasons and other potential public health threats, thereby safeguarding the most vulnerable populations.

This study provides a nuanced understanding of the evolving trends in influenza vaccination uptake in the Spanish adult population across three different influenza seasons, particularly during and after the COVID-19 pandemic. By analyzing these changing attitudes and behaviors, this study contributes to existing practices by identifying specific shifts in public perceptions that are critical for tailoring future interventions. For instance, understanding the emergence of vaccine fatigue as a significant barrier highlights the need to refine communication strategies, focusing not only on the benefits of influenza vaccination but also on addressing the burnout associated with repeated public health campaigns. Additionally, this study emphasizes the importance of re-engaging healthcare professionals as active promoters of vaccination, as their influence remains one of the strongest predictors of vaccine uptake. These insights can help refine current public health approaches by targeting specific barriers that have emerged or intensified due to the pandemic context, thereby making vaccination campaigns more effective and resilient in the face of fluctuating public attitudes.

## Figures and Tables

**Figure 1 vaccines-12-01162-f001:**
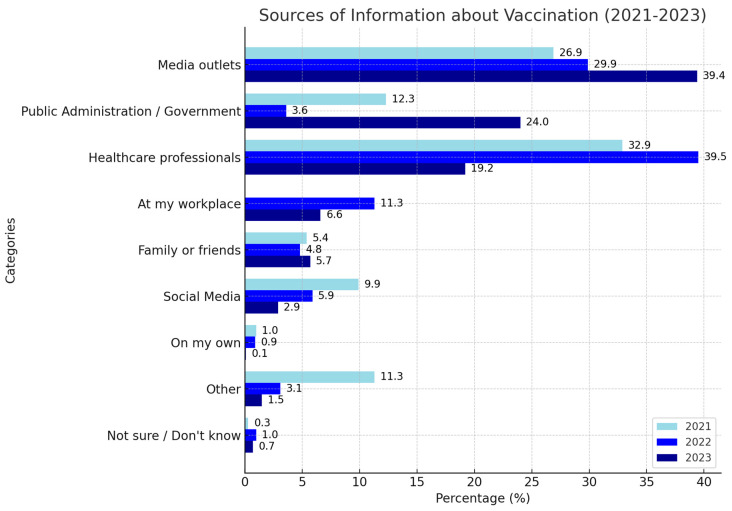
Primary sources of information on influenza vaccination for Spanish population.

**Table 1 vaccines-12-01162-t001:** Sociodemographic characteristics of study participants.

Variables	Survey 2023 N (%)	Survey 2022 N (%)	Survey 2021 N (%)
**Gender**			
Women	1154 (52.3)	1156 (52.1)	1139 (51.7)
Men	1052 (47.7)	1063 (47.9)	1066 (48.3)
**Age group**			
18–29	319 (14.5)	334 (15.0)	299 (13.6)
30–44	533 (24.2)	537 (24.2)	507 (23.0)
45–64	771 (34.9)	774 (34.9)	803 (36.4)
65 or more	583 (26.4)	575 (25.9)	596 (27.0)
**Employment Status**			
Private sector worker	749 (33.9)	656 (29.5)	659 (29.9)
Public sector worker	304 (13.8)	301 (13.6)	242 (11.0)
Self-employed/Business owner	159 (7.2)	151 (6.8)	131 (5.9)
Retired	607 (27.5)	620 (27.9)	669 (30.4)
Unemployed	170 (7.7)	275 (12.4)	268 (12.1)
Student	83 (3.8)	72 (3.2)	94 (4.3)
Homemaker	135 (6.1)	146 (6.6)	142 (6.4)
**Education Level**			
Primary or Basic	556 (25.2)	615 (27.7)	710 (32.2)
Secondary ^1^	783 (35.5)	932 (42.0)	869 (39.4)
University	867 (39.3)	672 (30.3)	626 (28.4)
**Belonging to Risk Groups**			
People over 60 years old ^2^	445 (60.3)		
Respiratory pathologies	139 (18.9)	125 (19.4)	115 (17.1)
Cardiac pathologies	65 (8.8)	80 (12.5)	79 (11.8)
Other chronic or immunosuppressed pathologies	73 (9.9)	71 (11.1)	66 (9.8)
Diabetics	59 (8.0)	57 (8.9)	19 (2.9)
Healthcare professionals	41 (5.5)	43 (6.7)	87 (12.9)
Immunosuppression	20 (2.7)	33 (5.1)	27 (4.0)
Essential workers ^3^	15 (2.1)	34 (5.4)	17 (2.5)
Other ^4^	18 (2.0)	96 (14.9)	14 (2.1)

^1^ Secondary education in Spain consists of Compulsory Secondary Education and Baccalaureate or Vocational Training degrees. ^2^ In July 2023, the Interterritorial Council of the National Health System published its recommendations for influenza and COVID-19 vaccination for the 2023–2024 season in Spain. In the age group recommendations, the indicated age was lowered from 65 to 60 years [34]. ^3^ Essential workers are considered to be those indicated by the government Royal Decree-Law 10/2020 of 29 March published by the Government of Spain [35]. ^4^ Within the category “other”, there are three risk groups: people with disabilities, pregnant women, and people living with at-risk patients.

**Table 2 vaccines-12-01162-t002:** Attitudes and practices in relation to previous influenza vaccination level.

Variable	Values	Annual or Frequent or Sporadic Vaccinations ^1^	
		N/Total (%)	*p*
Risk group	Yes	591/739 (80.0)	**<0.001**
	No	536/1459 (36.7)	
Age	>65	491/582 (84.4)	**<0.001**
	<65	641/1615 (39.7)	

^1^ Annually (once a year), frequently (once every two years), sporadically (once every two years or more). The *p*-values highlighted in bold indicate that they are statistically significant results (<0.05).

**Table 3 vaccines-12-01162-t003:** Reasons why people chose to get influenza vaccinated or not in previous campaigns.

	Reasons	Survey 2021 N (%)	Survey 2022 N (%)	Survey 2023 N (%); *p*
Reasons for deciding to vaccinate	For my own protection and/or environment	755 (71.5)	770 (71.7)	703 (62.1); **<0.001**
	Because my doctor recommended it	642 (61.0)	633 (59.3)	649 (57.3); 0.871
	Because of social responsibility	296 (28.1)	319 (30.4)	219 (19.3); **<0.001**
	Because I have suffered from the consequences of influenza in the past	180 (17.1)	181 (17.4)	72 (6.4); **<0.001**
	Because I have sufficient information about the vaccine	176 (16.7)	220 (21.1)	35 (3.1); **<0.001**
	Because it was recommended to me by nurses	130 (12.3)	128 (12.3)	61 (5.4); **<0.001**
	To get vaccinated simultaneously against influenza and SARS-CoV-2		204 (19.7)	110 (9.7); **<0.001**
Reasons for deciding not to vaccinate	Not recommended/prescribed	437 (38.6)	575 (51.8)	712 (66.3); **<0.001**
	My doctor has not recommended it	411 (36.3)		
	I do not consider influenza to be serious	345 (30.5)	335 (32.0)	420 (39.1); **<0.001**
	I don’t have enough information about the vaccine	115 (10.2)	156 (14.9)	64 (6.0); **<0.001**
	Lack of confidence in the effectiveness of the vaccine	93 (8.2)	112 (10.7)	91 (8.5); 0.103
	I have had a previous vaccination and it made me feel sick	78 (6.9)	73 (6.9)	49 (4.5); **0.024**
	I have a phobia of needles	42 (3.7)	57 (5.4)	23 (2.1); **<0.001**

The *p*-values highlighted in bold indicate that they are statistically significant results (<0.05).

**Table 4 vaccines-12-01162-t004:** Attitudes and behaviors regarding intention to get influenza vaccinated in the 2023/24 campaign.

Variable	Values	Intention to Vaccinate in the 2023/24 Campaign	
		N/Total (%)	*p*
Risk group	Yes	573/706 (81.2)	**<0.001**
	No	401/1409 (28.7)	
Age	>65	456/566 (80.6)	**<0.001**
	<65	524/1557 (33.7)	
Vaccination in recent years	Yes, annually	741/759 (97.6)	**<0.001**
	Yes, with some frequency	62/90 (68.9)	
	Yes, sporadically	99/284 (34.9)	
	No	75/1065 (7)	
Employment status	Private sector worker	218/720 (30.3)	**<0.001**
	Public sector worker	126/282 (44.7)	
	Self-employed/entrepreneur	35/152 (23.0)	
	Retired or pensioner	455/586 (77.6)	
	Unemployed	54/169 (32.0)	
	Student	16/82 (19.5)	
	Unpaid domestic work	76/130 (58.5)	
Level of studies	Primary or lower	324/528 (61.4)	**<0.001**
	Secondary	318/760 (41.8)	
	University	338/835 (40.5)	

The *p*-values highlighted in bold indicate that they are statistically significant results (<0.05).

**Table 5 vaccines-12-01162-t005:** Reasons why people expressed that they were hesitant to get influenza vaccinated or not in the 2021/22, 2022/23 and 2023/24 campaigns.

	Reasons	Survey 2021 N (%)	Survey 2022 N (%)	Survey 2023 N (%); *p*
Reasons for deciding to vaccinate	You need to protect yourself against viruses	748 (67.4)	734 (70.0)	561 (57.3); **<0.001**
	It is necessary to get an annual influenza vaccination	596 (53.8)	567 (55.5)	483 (49.3); **0.006**
	I trust vaccines in general	403 (36.7)	435 (42.8)	264 (27.0); **<0.001**
	Because of social responsibility	358 (32.5)	444 (43.8)	346 (35.3); **<0.001**
	Because COVID-19 has made me more aware of the importance of vaccination.	239 (21.7)	257 (25.4)	58 (5.9); **<0.001**
	In case I have COVID-19, it will help me with the effects of the vaccine.	212 (20.0)		
	To be vaccinated against influenza and COVID-19 at the same time		270 (27.4)	66 (6.8); **<0.001**
Reasons for deciding not to vaccinate	Not recommended/prescribed	456 (42.6)	552 (49.9)	672 (54.8); **0.018**
	No vaccination is necessary	335 (31.7)	288 (27.8)	300 (24.5); 0.072
	I do not consider influenza to be a serious (deadly) virus	246 (23.2)	315 (29.7)	392 (32.0); 0.238
	I do not trust vaccines in general	116 (10.9)	155 (14.8)	98 (8.0); **<0.001**
	COVID-19 vaccine is sufficient	112 (10.7)	80 (7.7)	25 (2.0); **<0.001**
	Not effective	96 (9.1)	120 (11.4)	42 (3.4); **<0.001**
	I got vaccinated once, and it made me feel sick			94 (7.7)

The *p*-values highlighted in bold indicate that they are statistically significant results (<0.05).

**Table 6 vaccines-12-01162-t006:** Predictors of vaccination intention for the 2023/24 campaign.

Variables	Categories	*p*	OR	95% CI
Age	>65	**<0.001**	2.0	1.4–2.8
	<65		1	
Risk group membership	Yes	**<0.001**	5.4	3.9–7.5
	No		1	
Previous vaccination	Yes	**<0.001**	41.4	30.7–55.9
	No		1	

The *p*-values highlighted in bold indicate that they are statistically significant results (<0.05).

**Table 7 vaccines-12-01162-t007:** Measures to encourage influenza vaccination.

Measures	Survey 2021 a Lot + Quite N (%)	Survey 2022 a Lot + Quite N (%)	Survey 2023 a Lot + Quite N (%); *p*
Having it recommended by my doctor, nurse, or pharmacist			935 (85.8)
Having more information about vaccination and its benefits	897 (81.1)	1771 (79.8)	906 (81.2); 0.216
Facilitating access to Primary Care Centers	812 (73.9)	1701 (76.6)	
A media awareness campaign	746 (67.9)	1570 (70.7)	681 (62.4); **<0.001**
Sending annual reminders via SMS	775 (70.1)	1474 (66.4)	758 (67.9); 0.238
Joint vaccination with other vaccines, such as pneumococcal or COVID-19			662 (59.3)
People in my environment deciding to get vaccinated	586 (53.4)	1259 (56.7)	611 (56.1); 0.492
Creation of mass vaccination sites like those for COVID-19.	690 (62.3)	1241 (55.9)	575 (52.8); 0.302

The *p*-values highlighted in bold indicate that they are statistically significant results (<0.05).

## Data Availability

The data presented in this study are available upon request from the corresponding author. The data are not publicly available due to the privacy policies of the company that carried out the surveys.

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
