# Peer review of "Analyzing Changes in Attitudes and Behaviors towards Seasonal Influenza Vaccination in Spain’s Adult Population over Three Seasons"

_vaccines, 2024, doi:10.3390/vaccines12101162_

Round 1

Reviewer 1 Report

Comments and Suggestions for Authors

In my opinion, the paper is clear, readable or informative and will provide a valuable source document for anyone who needs a beginner to know and understand this issue.    

The introduction is relevant and grounded in theory. Sufficient information on the findings of previous studies is presented to enable readers to follow the rationale and procedures of this study. The knowledge gap is generally well delineated. The aim of this study is well expressed and of interest.  

Overall, the article follows a proper structure. 

The appropriate methodology was applied in the work.

The results are presented and described clearly and in detail.

In the Discussion section, the results of this study are appropriately compared with the results of similar studies.     

The research appears to have been appropriately conducted in accordance with ethical standards.

The conclusions are generally supported by the results.

Minor comment: at the end of the Discussion section add a paragraph in which the limitations of this study should be discussed.   

Reviewer 2 Report

Comments and Suggestions for Authors

Dear Authors

The subject of this paper is very interesting and above all very relevant.

Studying the follow-up of vaccines is fundamental to avoid epidemics and pandemics.

Taking advantage of the fact that the Pisuerga passes through Valladolid, I would like to make some comments with the intention of improving the paper.

It is said that a non-probabilistic sample was taken, but this sample was selected by simple random sampling.

A sample of 2206 people was taken. Non-probability sampling does not require sample size calculation, but it also does not require such a large sample and is usually done with convenience.

Making inference with this type of sampling can cause biases that I have not seen in the limitations section.

You say that you do non-probabilistic sampling by quotas.

I believe that you should have done a stratified sampling (age, sex, etc.) and by clusters (geographic region).

I ask you to be clearer in this section of the sample.

Were participants asked for informed consent?

In addition, contact with participants was made by telephone calls to landlines and cell phones. This may produce selection bias and should be put into limitations and how to resolve this bias.

The first paragraph of the Data Analysis section seems to correspond to the analysis of the sample size. This is in contradiction with non-probability sampling and is in agreement with what I have just stated.

In this same paragraph I would like to know on which variable P=Q=0.5 is taken and why it is used as precision +- 2.1%. I assume it will be from the vaccination variable (yes/no), but it should be said.

They state that the methodology is sound and in paragraph 2 of Data Analysis they say that a rigorous statistical analysis has been performed to facilitate the identification of patterns. However, they do not say what statistical analysis they are referring to.

Then they mention logistic regression to obtain odd ratios. This is correct.

I suggest deleting the last sentence in the Data Analysis section because these statistical techniques and regression models are widely used.

I would like to know what statistical analysis was done in Table 2. I assume the Z-test for differences of proportions or the Chi-Square test. It should be clarified in the data analysis section.

I also can't figure out what statistical analysis was done for Tables 3, 4 and 5. Where do those p-values come from.

In table 6 I assume that the odd ratio adjusted for the variables chosen is shown. If so it should be said in the data analysis section.

Therefore, I cannot know if the results are correct without knowing what analysis has been performed.

If the results are not correct, the conclusions obtained may not be correct.

Reviewer 3 Report

Comments and Suggestions for Authors

1. The manuscript presents the conclusions of the study already in the Introduction, which is not a sound logic of writing. The purpose of the study is not inverted from the findings of the study, but is presented without knowing the findings of the study yet.

2. Although the study explored trends of change, its predictors were not new. I believe that readers would have preferred to see if different predictors would have been created after being influenced by COVID-19.

3 Although this study emphasizes the urgency and significance of some practice initiatives, it is not significantly different from some of the existing calls. Please further describe how this study makes an additional contribution to existing practices? For example, how can an understanding of developing trends help to further refine the process of guiding practice?

4. Attitudes, acceptances and intentions are different concepts. Please take care to distinguish between the concepts and how they are measured.

5. The discussion section of this manuscript is too cluttered. It is more like a research report than an academic paper. Please present the highlights of this study in the form of a dialogue with the existing literature rather than analyzing the findings of this study alone.

6. What are some of the issues raised by this study that have already been identified in practice? What are some that have not yet been identified? Please further highlight the value of this study in context.

7. Please add a discussion of the limitations of this study and the outlook for future research.

8. Due to the fact that different groups are researched each year, this study needs to further illustrate through statistical analysis that there is no significant difference in the structure of the research groups in each round of research.

Round 2

Reviewer 2 Report

Comments and Suggestions for Authors

The authors have clarified all my doubts and have responded to all my comments
Thank you very much.

Reviewer 3 Report

Comments and Suggestions for Authors

Thanks for the revisions. I agree to accept this version.